# Interplay between Neutrophils, NETs and T-Cells in SARS-CoV-2 Infection—A Missing Piece of the Puzzle in the COVID-19 Pathogenesis?

**DOI:** 10.3390/cells10071817

**Published:** 2021-07-19

**Authors:** Paulina Niedźwiedzka-Rystwej, Ewelina Grywalska, Rafał Hrynkiewicz, Dominika Bębnowska, Mikołaj Wołącewicz, Adam Majchrzak, Miłosz Parczewski

**Affiliations:** 1Institute of Biology, University of Szczecin, 71-412 Szczecin, Poland; rafal.hrynkiewicz@usz.edu.pl (R.H.); dominika.bebnowska@usz.edu.pl (D.B.); 2Department of Clinical Immunology and Immunotherapy, Medical University of Lublin, 20-093 Lublin, Poland; ewelina.grywalska@gmail.com; 3Department of Environmental Microbiology and Biotechnology, University of Warsaw, 02-096 Warsaw, Poland; mikolaj.wolacewicz@gmail.com; 4Department of Pediatric Infectious Diseases, Independent Public Regional Hospital in Szczecin, 71-455 Szczecin, Poland; adammajchrzak@protonmail.com; 5Department of Infectious, Tropical Diseases and Immune Deficiency, Pomeranian Medical University in Szczecin, 71-455 Szczecin, Poland; mparczewski@yahoo.co.uk

**Keywords:** COVID-19, immunity, SARS-CoV-2, NET, neutrophil, T cells

## Abstract

Since the end of 2019, a new, dangerous virus has caused the deaths of more than 3 million people. Efforts to fight the disease remain multifaceted and include prophylactic strategies (vaccines), the development of antiviral drugs targeting replication, and the mitigation of the damage associated with exacerbated immune responses (e.g., interleukin-6-receptor inhibitors). However, numerous uncertainties remain, making it difficult to lower the mortality rate, especially among critically ill patients. While looking for a new means of understanding the pathomechanisms of the disease, we asked a question—is our immunity key to resolving these uncertainties? In this review, we attempt to answer this question, and summarize, interpret, and discuss the available knowledge concerning the interplay between neutrophils, neutrophil extracellular traps (NETs), and T-cells in COVID-19. These are considered to be the first line of defense against pathogens and, thus, we chose to emphasize their role in SARS-CoV-2 infection. Although immunologic alterations are the subject of constant research, they are poorly understood and often underestimated. This review provides background information for the expansion of research on the novel, immunity-oriented approach to diagnostic and treatment possibilities.

## 1. Introduction

At the end of 2019, the world unexpectedly faced a new challenge related to the appearance of the new Severe Acute Respiratory Syndrome Coronavirus 2 (SARS CoV-2), the agent causing the current coronavirus disease 19 (COVID-19) [1]. Previously identified coronaviruses, SARS-CoV and MERS-CoV (Middle East Coronavirus), have been historically responsible for acute respiratory distress syndrome (ARDS), which is characterized by widespread inflammation in the lungs as a result of an abnormal inflammatory response to viral infection [2]. However, outbreaks of these two CoVs have been limited in size and, despite significant mortality, have largely been contained [3]. Therefore, despite the fact that SARS-CoV and MERS-CoV posed a significant threat to public health, albeit with limited impact, SARS-CoV-2 appears to be the most dangerous pandemic faced in the modern era.

The key elements to combat such a deadly phenomenon as the COVID-19 pandemic are quick diagnosis of the infection, effective treatment methods and, in particular, the implementation of safe and effective preventive measures. These measures include frequent washing and disinfecting of hands, wearing face masks, covering the mouth and nose when sneezing, maintaining social distancing, avoiding crowds and poorly ventilated spaces, frequent disinfection of rooms and everyday appliances, and preventive vaccinations [4]. However, the aim of vaccination is not only to protect the vaccinated people, but also to allow the acquisition of herd immunity and significantly reduce the transmission of the virus. This will result in minimizing the number of patients and extinguishing the pandemic. Understanding the pathogenesis of COVID-19, in addition to the importance of the immune system responses to the disease, are essential elements in research on developing and improving the above-mentioned factors in the fight against pandemics in practice. The role of the immune system in SARS-CoV-2 infection is described by many authors [2,3,5,6] and most reviews focus on similar elements. In contrast, the current review emphasizes the rarely described elements of the immune system that may be incorporated in COVID-19 pathogenesis. 

It is widely known that the presentation of viral antigens to T and B cells takes place via the major histocompatibility complex (MHC) on antigen presenting cells (APCs) and thus activates the mechanisms of innate and adaptive immunity. It was reported that SARS-CoV-2 acts as an antagonist against type I interferons, which inhibits innate mechanisms [7]. Additionally, SARS-CoV-2 down-regulates MHC class I and II molecules, which affects the impairment of the immune response with the participation of T lymphocytes [5]. Then, as the disease progresses, the number of lymphocytes decreases with uncontrolled release of inflammatory cytokines—a cytokine storm [8]. This phenomenon is the main cause of ARDS, causing lung damage and multiple organ failure [9]. In this paper, we wish to highlight elements of the immune system that are less frequently mentioned in the context of COVID-19, which we believe may be key to a better understanding of the concept of immunity in SARS-CoV-2 infection. In this context, the role of neutrophils, which constitute an important element of innate immunity, as one of the elements of the first line of defense against microbes, remains underestimated by many authors. In addition to their ability to phagocytose, the formation of neutrophil extracellular traps (NETs), in which nuclear fragments and granules are actively released, is significant [9]. The role of these cells is even more important, as altered neutrophil numbers are recorded in the pathophysiology of the severe course of COVID-19 [10]. The significant role of macrophages in the pathogenesis of COVID-19 has also been described [11,12]. Feng et al. [13] reported that CD169+ macrophages in the spleen and lymph nodes can contribute to virus replication and spread. Moreover, a cytokine storm is a characteristic feature of the macrophage activation syndrome (MAS), i.e., massive macrophage activation, which plays a major role in ARDS [14]. In MAS, the innate immune response and IL-18 affect cytotoxic T cells (CTLs), and influence their interaction with antigen presenting cells (APCs) and their permeation with macrophages. Increased innate immune response and insufficient cytolytic removal of activated cells results in permanent production of inflammatory mediators, enhanced CTL-APC interaction, and a cytokine storm [11]. The pathogenesis of ARDS indicates the participation of pulmonary alveolar (AM) and interstitial (IM) macrophages, and the course of the disease is influenced by the time and degree of macrophage polarization. Zhang et al. described [15] that the cytokine storm in the severe course of SARS-CoV-2 infection begins in the lungs and is associated with an increased number of pro-inflammatory macrophages and the occurrence of pyroptosis in lung tissue, which promotes the release of a large number of pro-inflammatory cytokines. In patients with a severe course of COVID-19, in addition to hypercytokinemia, high levels of ferritin, CRP, and D-dimer were also recorded, which indicated the development of severe MAS inflammation and fibrinolysis [16]. Clinical trials describing the involvement of macrophages in ARDS have been conducted. In their research, Huang et al. [17] conducted an analysis of patients with severe COVID-19, which showed that the number of activated monocytes/macrophages was significantly increased, despite the fact that blood counts showed normal values of these cells. In turn, Zhou et al. [18] showed that in patients with ARDS, the number of CD16+ monocytes, including the non-classical CD14+ CD16++ population and the CD14++ CD16+ pro-inflammatory intermediate, was significantly increased. Moreover, the relative number of CD14++ CD16+ monocytes comprised up to 45% of all monocytes. By comparison, other studies by Niles et al. [19] showed that myeloid dendritic cells (mDCs), and neither M1 pro-inflammatory macrophages nor M2 anti-inflammatory macrophages, were stimulated to produce pro-inflammatory cytokines in SARS-COV-2 infection. Many differences have also been described between the severe course of COVID-19 and typical MAS. First, activation of pulmonary macrophages in SARS-CoV-2 infection often occurs without other features of MAS. Second, the condition is associated with pulmonary intravascular coagulopathy (PIC), but not disseminated intravascular coagulation (DIC) [16]. Moreover, hyperferritinemia and high CRP levels in COVID-19 are lower than the values observed during MAS [20]. Moreover, it has been described that the cytokine storm in SARS-COV-2 infection is characterized by a slight increase in key cytokines [16]. Due to these data, the status of monocyte/macrophage involvement in the pathogenesis of COVID-19 requires further consideration. In the course of SARS-CoV-2 infection, lymphopenia is also observed, which correlates with the severity of the disease [21]. It has been reported that lymphopenia may be associated with depletion of CD8+ T lymphocytes, but also the activity of CD4+ T lymphocytes, B lymphocytes, and NK cells, leading to a significant immunopathology [22,23]. As the search for prognostic markers for the course of the disease and response to treatment is vital, particular attention should be paid to the importance of neutrophil function, NETs, and lymphocyte exhaustion. In this review, we focus on information that we believe is key to better understanding the status of the immune system in SARS-CoV-2 infection, which may aid the development of more effective diagnostic, therapeutic, and prophylactic methods in the future.

## 2. Origin of SARS-CoV-2

The SARS-CoV-2 virus is the seventh coronavirus discovered to infect humans. It was first identified in Wuhan, Hubei province, China [1,17,18,24]. SARS-CoV, MERS CoV, and SARS-CoV-2 can each cause serious diseases, especially of the respiratory system, whereas HKU1, NL63, OC43, and 229E viruses are mainly associated with a mild course and yearly recurrence of the infection [25,26].

According to data provided by the World Health Organization (WHO), as of 28 June 2021, 122,271,944 cases and 2,700,669 deaths have been recorded due to COVID-19, and the virus has spread to 223 countries and territories (Coronavirus disease (COVID-19) (who.int)) [27].

Scientists are particularly interested in the origins of the virus [25]. This information is not only important from an epidemiological perspective. A detailed understanding of how the virus crossed the species boundary could help inhibit the spread of other new viruses in the future and ultimately prevent global pandemics. In addition, information about the origin of the virus appears to be of interest to the public. Research by Andersen et al. [25], i.e., a comparative analysis of the viral genomes (human SARS-CoV and SARS-CoV-2 viruses, the virus found in bats and pangolins, and also Bat-SARS-CoV-related viruses), clearly showed that SARS-CoV-2 is not a deliberately manipulated virus. Nonetheless, it is not currently possible to formulate a precise and unequivocal theory about its origin or the timeline of its zoonotic transmission [25]. It is generally agreed that SARS-VoV-2 originated naturally from bat coronaviruses (probably RaTG13). However, the proximal origin of the virus is questionable, due to significant discrepancies of the adaptation patterns of SARS-CoV-2 and other coronaviruses [28].

### 2.1. Molecular and Structural Characteristics of SARS-CoV-2

The SARS-CoV-2 genome consists of positive-sense single-stranded RNA [24,26]. Its size is approximately 29.9 kbp (NC_045512.2) [29]. As a new *betacoronavirus*, it shows 79% sequence similarity with the SARS-CoV virus and 50% with the MERS-CoV virus [30]. The SARS-CoV-2 genome consists of 13–15 (12 functional) open reading frames (ORFs). The percentage of GC pairs in the SARS-CoV-2 genome is 38%. Eleven protein-coding genes have been identified in viral RNA, including 12 expressed proteins [26]. The ORF distribution of this virus closely resembles that of the SARS-CoV and MERS-CoV viruses [31,32]. ORFs are arranged in order: replicase and protease (1a–1b), and the structural proteins: spike glycoprotein (S), envelope protein (E), membrane protein (M), and nucleoprotein (N), which have the typical 5′-3′ appearance and are considered the primary targets of drugs and vaccines. These gene products play an important role in the entry, fusion, and survival of the virus in the host cells [26,33].

The main protein that plays a role in pathogenesis is the S protein (spike glycoprotein), which binds to the cell through the receptor-binding domain (RBD) [34,35]. The S protein causes the virion to fuse to the host cell [26]. This protein is composed of 1273 amino acids and consists of two subunits: (i) S1—responsible for the attachment of virions to the host cell membrane through interaction with the human ACE2 receptor, which initiates the infection process [26,35,36]; and (ii) S2, which acts as a fusion protein, helping the virion to bind to the mammalian cell membrane [26].

Other proteins of importance are the envelope proteins (E proteins). These are small proteins, composed of 75 amino acids, which play an important role in the assembly and release of virions [37,38].

The nucleocapsid proteins (N proteins) are proteins that play an important role in packaging viral RNA into the nucleocapsid [39]. This mediates the formation of viral particles by interacting with the viral nucleic acid and the M protein, which are helpful in enhancing the transcription and replication of viral RNA [26,40]. N proteins are also highly conserved within coronaviruses. The SARS-CoV-2 N protein shows 90% sequence similarity to the N protein of the SARS-CoV virus [26]. Therefore, N proteins represent a potential target of anti-CoV drugs.

Membrane proteins (M proteins) are structural proteins that function with all other viral proteins, in particular the E, N, and S proteins, and play an important role in RNA packaging. They have a size of 222 amino acids and are the most abundant proteins in the coronavirus family, giving them their characteristic shape [38].

An important protein that helps in the cleavage of the host RNA and replication of the viral RNA is the enzymatic protein, replicase polyprotein [41]. Replicase polyproteins are multifunctional proteins that perform various tasks contributing to the pathogenesis of viruses [42]. However, the main role of these proteins is to aid the transcription and replication of viral RNA [26].

### 2.2. SARS-CoV-2 Mechanism of Action

As mentioned earlier, SARS-CoV-2 belongs to the group of viruses from the CoV family, which can cause severe respiratory diseases [25,26]. Viral particles enter the body through ACE2 receptors (angiotensin-converting enzyme 2). ACE2 belongs to the family of membrane-associated carboxydipeptidases and is widely distributed throughout the body. It is highly expressed not only in the lungs, but also in the heart tissue, small intestine, and kidneys [42]. ACE2 in the lung is concentrated mainly in type II alveolar cells and macrophages, and to a lesser extent in bronchial and tracheal epithelial cells.

In the case of the SARS-CoV, the virus particles enter the body through this receptor. The viral S glycoprotein binds to the ACE2 receptor, and the viral and cell membranes fuse. This causes the virus to cross the cell membrane barrier and allows viral replication in the host cells (Figure 1) [42,43,44].

Hassan et al. [45] conducted a comprehensive bioinformatics study that analyzed the ACE2 gene sequences of nineteen different species to determine the possibility of SARS-CoV-2 transmission between many different species and humans. As a result, the high probability of interspecies viral transmission was confirmed, because in the analyzed amino acids of the ACE2 protein they showed similar binding properties with the receptor binding domain of the viral spike protein S. The results indicate that among the studied species, primates are the most endangered by interspecific transmission, followed by carnivores, cetartiodactyls, and, finally, bats [45].

Virus replication results in a negative regulation of ACE2, which in turn leads to the degradation of angiotensin II and the production of angiotensins 1–7. This subsequently activates the masoncogene receptor, which is responsible for the negative regulation of angiotensin II, mediated by the type 1 angiotensin II receptor (AT1R) [42,43,44]. Activation of AT1R ultimately leads to acute lung damage. The mechanism of action of the SARS-CoV-2 virus is presumed to be identical to that seen in ARDS [42,46] 

## 3. Innate Immunity—The Role of Neutrophils and NETs in COVID-19

Innate immunity is the body’s primary response to infection and plays a significant role in the pathogenesis of COVID-19. It has been suggested that innate immunity contributes to the development of cytokine storms and progression to severe disease [47]. For this reason, characterizing the elements and phenomena of innate immunity, which are highly distinctive in patients with SARS CoV-2 infection, is an essential element in developing new therapeutic approaches and fully understanding the pathogenesis of the disease. 

It has been observed that SARS-CoV-2 triggers innate immune responses via PAMP (pattern associated molecular patterns) and sensing PRRs (pathogen recognition receptors), including TLRs (Toll-like receptors), RIGs (retinoic acid-inducible genes), and MDAs (melanoma differentiation-associated proteins) [48]. Because local tissue damage in lungs is one of the main characteristics of COVID-19, the release of DAMPs (danger-associated molecular patters) has also been registered [48]. This leads to antiviral response via the activation of IFN types 1 and 3, upregulation of IL-6 and IL-1β, and further recruitment of neutrophils, which are our main point of interest.

Neutrophils (PMN cells, polymorphonuclear leukocytes) are components of innate immunity and a vital element of the body’s first line of defense. They are also involved in the pathogenesis of COVID-19 [49] because these cells are first recruited to the site of inflammation by a chemotactic signal from damaged lung cells. Histopathological findings from patient files showed the presence of neutrophil infiltrates in the lung capillaries, their extravasation into the alveolar spaces, and neutrophilic mucositis. Excessive infiltration of neutrophils at the site of infection may evolve into significant dangerous consequences. Due to the intensive degranulation of cells and the release of NET networks, cytokines and chemokines are overexpressed, which intensifies the inflammation, and may lead to a phenomenon of a cytokine storm, which poses a threat to the patient’s life [50]. Moreover, during COVID-19, an increased number of neutrophils with concomitant lymphopenia is observed. It has been confirmed that high neutrophil to lymphocyte ratio (NLR) indices are a harbinger of the severe course of the disease and correlate with the risk of death in patients. Therefore, the determination of NLR can be treated as a prognostic indicator in the early stage of the disease [51,52,53]. The aim of this work is to characterize the contribution of neutrophils and, in particular, the NETs produced by these cells in COVID-19, to improve understanding of the disease and its interaction with the immune system.

### 3.1. Neutrophils in SARS-CoV-2 Infection

Neutrophils, polymorphonuclear leukocytes (PMNs), belong to the most numerous groups of cells of the immune system [54,55,56]. In humans, they constitute approximately 60–70% of all peripheral blood leukocytes [57]. They arise because of the process of granulocytopoiesis in the bone marrow. Certain transcription factors are required for neutrophil maturation during granulopoiesis at a steady state, including PU.1, CCAAT/enhancer binding protein α (C/EBPα), growth factor independence 1 (GFI1), and C/EBPε [58].

From the bone marrow, PMN cells migrate to the circulatory system, where they live for about 7–10 h, after which they usually die by apoptosis. In the physiological state, the adult bone marrow can produce 5–10 × 10^10^ new neutrophils daily [59,60]. Neutrophils are the body’s first line of defense. They are recruited from immune cells to the site of injury within minutes after injury and are a hallmark of acute inflammation [56]. Factors that cause PMN migration may be factors of bacterial origin, e.g., bacterial formyl peptides (fMLP), factors secreted after tissue damage, e.g., mitochondria of dying cells, or factors secreted by neutrophils themselves or other cells involved in an inflammatory reaction, e.g., leukotriene B4 (LTB4) or chemokines (e.g., interleukin 8, IL-8) [58,61]. Neutrophils play an important role in the fight against both bacterial and fungal infections. Although the role of PMN cells in bacterial and fungal infections is well understood, their influence on antiviral activity has not yet been fully elucidated [54,55,56,62]. As neutrophils are the body’s first response to viral infection, their populations expand in the local microenvironment following viral infection [62]. Presence of pathogens in the organism and/or damage to the host tissues are the factors that initiate the inflammatory reaction [58]. In response to invasion by microorganisms, local macrophages and mast cells secrete TNF, IL-1β, and several other cytokines that activate endothelial cells that then capture circulating neutrophils [62]. PMN cells are present in the lungs during acute respiratory distress syndrome (ARDS), which can be caused by a wide variety of pathogens, including an array of viruses (for example, in influenza and CoVs2 infections), in addition to trauma and autoimmune disorders. Studies suggest that neutrophil recruitment to the lungs is associated with disease exacerbation in the course of viral infections [63,64].

Previous studies on people infected with SARS-CoV-2 have shown that, in patients with a severe course of COVID-19, characteristic changes in the phenotype, functionality, and number of PMNs are observed. An increased number of neutrophils in the nasopharyngeal epithelium and in the lower sections of the lungs [9,10,65] is found in these patients. In pulmonary autopsies of patients who died from COVID-19, neutrophil infiltrations into the pulmonary capillaries with extravasation into the alveolar space and neutrophilic mucositis were observed, indicating complete lower respiratory tract inflammation [66]. In patients with confirmed severe SARS-CoV-2 infection, an immature phenotype and/or dysfunctional mature neutrophils were found [66,67,68]. Increased infiltration of immature and/or dysfunctional neutrophils in severe COVID-19 has been shown to significantly contribute to an imbalance in the lung immune response [67,68]. In the case of COVID-19, neutrophils may play a protective role, but the extensive and long-term activation of these cells may have an adverse effect on patients’ lungs, which may result in pneumonia and/or acute respiratory distress syndrome (ARDS) [69]. 

Bronchoalveolar lavage fluid (BALF) studies in SARS-CoV-2 infected patients showed an increase in chemokine expression (CXCL-2 and CXCL-8), which facilitates the recruitment of neutrophils to the site of infection [70,71,72,73]. The infiltration of the neutrophil into the SARS-CoV-2 infection gene results in an accumulation of granules, which may consequently influence the physiopathology of ARDS. In addition, the hypoxia significantly influences the release of a large amount of ROS in the form of superoxide radicals and hydrogen peroxide, which leads to oxidative stress, contributing to the formation of a cytokine storm and the formation of blood clots [49,74,75]. It has been observed that the excessive neutrophil concentration coincides with the progression of lung lesions in patients with severe COVID-19 [76]. Due to the increased degranulation of the primary granules and the increased release of pro-inflammatory cytokines, neutrophils significantly contribute to the maintenance of inflammation in the lungs, which may lead to severe damage to the alveolar tissue, regardless of the cytopathic effect of SARS-CoV-2 virus [68].

An increase in the number of PMN cells, in addition to the nasopharynx and lung epithelium, has also been observed in the blood. Following the increase in NLR, the levels of D-dimers and CRP protein increase significantly [52,77,78,79,80,81,82,83,84]. Increased NLR levels have also been recognized as an independent predictor of mortality in hospitalized patients with confirmed certain comorbidities, such as diabetes and cardiovascular disease [80,85,86]. It was observed that COVID-19 diabetic patients with a higher NLR showed a much more severe course and experienced a longer hospital stay [87].

### 3.2. Neutrophil Extracellular Traps (NETs)

One of the antimicrobial functions of neutrophils is to destroy pathogens through a special type of cell death called NETosis. During this phenomenon, chromatin decondensation from the neutrophil cell nucleus takes place, and then, after the nucleus membrane disintegrates, chromatin is mixed with proteins and granules, resulting in formation of neutrophil extracellular traps (NETs), which are included in the group of innate immunity mechanisms [88]. NETs are extracellular and three-dimensional lattices of decondensed chromatin and various nuclear and granular proteins, which primarily include histones (representing up to 70% of NET proteins), but also antimicrobial proteins from neutrophil granules, such as defensin, cathepsin, lactoferrin, and myeloperoxidase (MPO) [89,90]. NETs have strong antimicrobial properties, but have also been proven to play a key role in antiviral immunity by oxidative burst and phagocytosis [88,91,92,93,94]. Several enzymes are responsible for NET formation: neutrophil elastase (NE) (responsible for the degradation of intracellular proteins and the breakdown of the nucleus); peptidyl arginine deiminase type 4 (PAD4) (playing a role in decondensation and release of chromosomal DNA); and gasdermin D (its task is to create pores in the cell membrane to facilitate the expulsion of DNA and related molecules, including proteins, outside the cell) [95,96,97]. It is known that a viscous network of NETs can directly bind and immobilize viruses, thus mechanically preventing them from spreading and reaching the target cells. This is made possible by positively charged amino-acid-containing histone proteins, which can adhere to the negatively charged virus envelope [98]. Although NETs are beneficial in the host defense against pathogens, collateral damage from sustained NET formation also stimulates many disease processes, including those that occur during viral infections [92]. Neutrophil stimulation to produce NETs in response to contact with the virus is performed by PRR receptors, including TLR7 and TLR8 [62]. Furthermore, in the hantavirus model, NETs are induced by stimulation with the virus–β2 integrin receptor [99]. Then, due to the presence of MPO, cathelicidins, and α-defensin, the viruses are inactivated. Importantly, certain viruses have developed mechanisms to avoid NETs, which is possible due to the involvement of viral proteases capable of decomposing this network [89]. The immune function of NETs is not limited to direct antiviral action, because the components of NETs also act as DAMPs (danger-associated molecular patterns) and stimulate antiviral effector mechanisms of other immune cells, including the release of proinflammatory cytokines and chemokines. NETs can also activate dendritic plasmacytoid cells (pDCs) through TLRs, with an antiviral role associated with the release of type I IFN. NETs can also affect antiviral immunity by lowering the T lymphocyte activation threshold [92]. However, it should be mentioned that excessive production of NETs may also have negative effects; therefore, it is defined as a “double-edged sword” in immunity [55]. The over-formation of NETs can trigger a cascade of inflammatory reactions that promote cancer cell metastasis, destroy surrounding tissue, facilitate micro-thrombosis, and cause permanent damage to the pulmonary, cardiovascular, and renal organs. These are the most commonly affected and stressed organ systems in severe disease, including COVID-19 [20,93,100,101,102]. Long-term and extensive production of NETs is characterized by neutrophilic mucositis and leads to tissue damage and, thus, deterioration of health [103]. Moreover, the overproduction of NETs in viral respiratory tract infections may weaken lung epithelial barrier functions, but also NETs may directly lead to epithelial cell death. This has been confirmed by in vivo mouse studies, which showed acute lung damage and inflammatory response after influenza A virus provocation [91]. Studies have also been carried out [104] which showed the formation of NETs in asthma and chronic obstructive pulmonary disease (COPD) associated with immunopathology of the respiratory tract in these diseases, i.e., lung NETopathy [103]. Studies on RSV infection have shown that the fusion glycoprotein (protein F) induces the formation of NETs, and over-release of NETs occurs through the TLR4 pathway, which leads to exacerbation of inflammation in infected infants and young children [105].

### 3.3. Neutrophil Extracellular Traps (NETs) in COVID-19

Studies [93,106] have also shown that the severe course of COVID-19 is manifested by a marked increase in “low density granulocytes”, specific neutrophils also found in autoimmune diseases. Studies have confirmed a significant role of NETs in the pathogenesis of COVID-19-related respiratory distress syndrome (ARDS). Research has shown that NET levels are higher in patients with transfusion-related ARDS than in those without ARDS [93,106]. Moreover, neutrophils from patients with ARDS associated with pneumonia appear to be “triggered” to form NETs, and both the degree of excitation and levels of NETs in the blood correlate with disease severity and mortality [107,108,109].

The formation of NETs in COVID-19 is unregulated, and because it occurs across multiple sites, the lumen of the blood vessels in the lungs is quickly obstructed, which accelerates the incidence of ARDS [110]. It is worth adding that NET can activate neighboring macrophages to produce cytokines, including neutrophil stimulating IL-3β, which induces a coupled immune response that leads to a vicious cycle of aggravating inflammation; this can simply be called the IL-3β-NET loop [111]. NET involvement in ARDS was demonstrated by postmortem studies of the lung tissues of COVID-19 patients, which showed that microvascular thrombi of NET-releasing neutrophils were mixed with platelets. Moreover, in patients included in these analyses, an acute course of the disease and sudden death was observed [112]. A significant proportion of severe COVID-19 patients are elderly with defective NET production, NET efferocytosis, and necrotic debris. Such a state leads to a significant difficulty in extinguishing very severe inflammation, and thus contributes to the deterioration of health in COVID-19 patients [62]. Thus, the current knowledge indicates that the determination of NET indicators in patients may be a prognostic factor for the patient’s survival [91].

## 4. T Cell Responses—Are Fewer Emphasized Subsets of T Cells and T Exhaustion Crucial for the Course of COVID-19 Infection?

The adaptive immune response is activated following viral uptake and antigen processing by a range of APCs [48]. Analogically, as in several viral infections, the role of APCs is to present viral antigen to B cells (Figure 2). Next, the differentiation of B cells into antibody-producing plasma cells takes place, and the neutralizing antibodies (nAbs) bind to key viral proteins, such as the spike, and neutralize their activity. Overall, T cells are the key players in antiviral immunity, including SARS-CoV-2 infection. They effectively eliminate virus-infected cells, impact the innate antiviral responses and support B cells. However, it must also be remembered that there are also several different immunological antibody-related processes involved during COVID-19, such as antibody-dependent cellular cytotoxicity (ADCC), antibody-dependent cellular phagocytosis (ADCP), and antibody-dependent complement activation (ADCA) [48]. 

Several studies have confirmed that the severity of COVID-19 is correlated with the T CD3+ decrease [21,113,114,115]. Simultaneously, the percentage of naïve helper T cells is increased, and that of memory helper T cells is decreased in severe cases [116].

In addition, the decrease in CD8+ T cells (cytotoxic T cells, CTLs) has been recorded. This is pivotal because CTLs kill virally infected cells via the production of granzymes and perforin, and the expression of the Fas ligand (FasL), all of which mediate cellular apoptosis [117]. In addition, patients with severe COVID-19 also showed a reduction in total CD4+ T cells, CD8+ T cells, B cells, and NK cells compared to mild and/or moderate COVID-19 patients [118,119].

It is important to emphasize that the decrease in CTLs is recorded mainly within the CD8+ T cell compartment, but the reason for this remains uncertain [117]. The following possibilities have been enumerated: trafficking of CD8+ T cells into tissues with ongoing SARS-CoV-2 replication, increased elimination of this T cell’s subset during the infection, or pre-existing low levels of this cell line in a severe course of the infection [117].

In the dramatic decrease in the above-mentioned cell subsets, the involvement of IL-10, IL-6, and TNF-a is highly emphasized [21,120].

In general, T cell immune responses are considered to be highly specific and to play a vital role in strong antiviral responses. However, the scale of T cell responses as beneficial or harmful to COVID-19 patients remains unclear due to evidence pointing to suboptimal, dysfunctional, or excessive activity [121]. In addition to showing that COVID-19 has decreased levels of CD4+ and CD8+ T cells, it has also been shown that the levels of CD4 + T cells expressing CD38− and HLA-DR−, and CD8+ T cells and CD4+PD-1+ T cells, were higher in the proportion of patients with severe COVID-19 compared to healthy people [22,122]. Importantly, the functionality of memory CD8 + T cells and CD8+ PD-1+ CD38+ T cells in severe COVID-19 patients was demonstrated by high expression levels of perforin and granzyme B compared to normal lymphocytes [121].

Follicular helper T cells (Tfhs) and Th2 CD4+ T cells both provide support for B cell antibody production [120]. The absence of Bcl-6+ TFH has been reported in patients with SARS-CoV-2 infection, together with the absence of germinal centers [123]. These findings may serve as a background for impaired humoral immunity and low levels of somatic hypermutation in antibodies from convalescent COVID-19 patients [124,125]. Among COVID-19 patients with mild disease, elevated activity of activated Tfh in blood is seen, suggesting that activated Tfhs are correlated with a mild picture of the disease [126]. After infecting the host with SARS-CoV-2, the uptake of viral antigens is performed by DCs, which further present them on MHC-II to naïve CD4+ T cells and induce Tfh cell differentiation toward type 1 polarization. Stronger Tfh1 cell polarization results in lower COVID-19 severity [126]. Another subset of Tfh that may play a significant role in COVID-19 is cytotoxic Tfh, for which levels are elevated relatively early in hospitalized patients and are correlated with levels of SARS-CoV-2 spike protein-specific antibodies [127]. Moreover, severe COVID-19 patients are also characterized by reduced CXCR5 expression, indicating impaired crosstalk between T and B cells, which may result in dysregulated humoral immunity [122,128]. Taking the above into account, the limited durability of antibody responses in SARS-CoV-2 infection may rely on the T/B crosstalk interference, and this further explains why gaining herd immunity through natural infection may be difficult.

Th1 and Th17 CD4+ T cells are also thought to play a role in the inflammatory and antiviral responses (Figure 2), which has been observed in COVID-19 infected patients [129]. The possible role is to increase the inflammatory condition of critical patients, whereas in these patients the frequency of proinflammatory Th1 and Th17 cells was significantly reduced [129]. These T subsets may migrate from the blood to the lungs of severely infected patients, simultaneously resulting in the decrease in these subpopulations in peripheral blood. Next, in lungs they promote tissue damage via inappropriate neutrophil recruitment and apoptosis inhibition [130]. At the same time, the recruitment of Th1 to lungs may result in macrophage triggering, leading to activation of a cytokine storm [131].

Regulatory T cells possess an immunoregulatory role in SARS-CoV-2 infection due to the production of anti-inflammatory cytokines and contact-mediated cellular suppression. Levels of these T subset cells are also decreased in patients with both mild and severe forms of the infection [48,116,132]. An interesting observation was registered by Vick et al. [133], who confirmed an increased frequency of Tregs expressing activation and suppression markers CTLA-4, ICOS, Ki67, HLA-DR/CD38, and PD-1 in patients with respiratory viruses compared to healthy donors. Moreover, several phenotypes of Tregs have been checked and the study revealed the so-called “SARS-CoV-2-specific” Treg pattern, where CD45RA-CCR7− (effector memory phenotype) was decreased in SARS-CoV-2 samples, including a CD27+ CD28+ ICOS+ HLA-DR+ Ki67− and a CD27+ CD28+ ICOS+ HLA-DR-Ki67− subset. In contrast, a subset of CD4+ CD25+ CD127− Treg, that is, CD45RA− CCR7+ (central memory phenotype) that co-expresses CD27, CD28, Ki67, and HLA-DR, was significantly increased in the circulation of patients with SARS-CoV-2 [133]. This information is of a great value to efficiently diagnose and react in the case of SARS-CoV-2 infection. 

In addition, the dysregulation of effector T cells and accumulation of exhausted T cells is one of the key features of COVID-19 disease [134]. The phenomenon of the exhaustion of memory T cells is visible by overall lymphopenia and increased levels of T_ex_ [134]; however, the exhaustion of cytotoxic lymphocytes in the course of COVID-19 has not been reported, although functional exhaustion of cytotoxic lymphocytes (CTL) is often correlated with disease progression [135]. Nevertheless, it has been shown that the total number of CTLs significantly decreases in patients infected with SARS-CoV-2 and the level returns to normal after the infection, but with a reduced expression of NKG2A [135], which is known to be a novel inhibitory molecule on the immune-checkpoint blockade [136]. The role of this molecule has been confirmed in antiviral [137] and antitumor immunity [138].

Considering all of the available data, because CTL decrease is characteristic of an antiviral immunity breakdown [139], it may be assumed that this breakdown occurs at the early stage of the infection with SARS-CoV-2, resulting in the disease progression [135,139]. Therefore, avoiding exhaustion of CTLs at the early stage of SARS-CoV-2 infection, namely using NKG2A targeting, may lead to virus elimination [135].

## 5. The Crosstalk between Neutrophils, NETs, and T Cells in COVID-19—What Is the Link?

To the best of our knowledge, the link between the elements of innate immunity described in this paper in the course of SARS-CoV-2 infection has not been established in any research, although significant amounts of knowledge have been gathered regarding these separate mechanisms. Nevertheless, as a rule, neutrophils constitute the first line of defense against infections and one of their strategies is the release of NETs. It is also known that NETs can activate DCs and impact inflammatory responses. Moreover, there are also studies showing that NETs may prime T cells by reducing their activation threshold [140]. In addition to the widely known DC link between innate and adaptive immunity, this may be another connection between those two forms of pathogen killing strategies that is not yet fully confirmed in COVID-19. Further studies are needed to understand this mechanism, and we trust that the collection of facts in this review might be helpful.

## 6. Conclusions

We are now equipped with a very powerful weapon—vaccines—against a new enemy, i.e., COVID-19. However, despite the importance of vaccinations, another area must be investigated to ultimately control the ongoing infection. Understanding the pathogenesis of COVID-19, considering the undeniable importance of innate immunity, is a crucial step to resolving these issues.

After more than a year of this fight, we are able to fully describe the clinical course of the disease, and have developed helpful guidelines to combat the virus, for which the choice of a suitable drug depends on the stage of the disease. Identification of the differences in immunity that are responsible for different courses of the disease in severe and non-severe patients remains to be investigated.

We have gathered a significant amount of data concerning the main mechanisms of immune response to SARS-CoV-2 infection. In this review, we emphasize the importance of the currently known information regarding immunity, in comparison to the obvious known facts that existed at the beginning of the pandemic, that play a crucial role in the first line of defense against the virus (Table 1). According to the currently known information, the vision of the immunity mechanisms that play a role in in SARS-CoV-2 infection has evolved. As time passes, we have become sure that innate immunity takes the lead in the fight against this virus (Table 1). To emphasize these facts, in this paper we describe the role of neutrophils in COVID-19 and underline their unique ability to regulate cytokine activity and to destroy pathogens through neutrophil extracellular traps (NETs). Second, we describe mechanisms that lymphocytes use to fight infection. We also try to answer the question, does exhaustion of T cells explain the deterioration in some groups of patients? 

If we wish to reduce the mortality of the COVID-19 pandemic, we must be a step ahead of the virus. The best way to achieve this is to expand our knowledge about the mechanisms of immunity against SARS-CoV-2. 

Taking this into consideration may be a significant step to provide a better understanding of COVID-19 pathogenesis and, therefore, to more accurately predict the course of infection. The development of novel diagnostic and therapeutic tools would allow the study and reverse exhaustion of T cells at the early stage of infection. 

The immune response to SARS-CoV-2 is extremely complex. In addition to the main mechanisms of innate and acquired immunity, other factors that are not commonly known should also be emphasized. Immune checkpoint molecules are highly important proteins that control the physiological immune response and prevent over-activation of the immune system. Among these, the most comprehensively investigated are PD-1/PD-L1 and CTLA-4; however, other proteins, such as LAG-3, TIM-3, and TIGIT, should also be noted. The measurement of their upregulated expression on the surface of immune cells can be used as a predictive value in various conditions, such as cancers and chronic viral infections [141], and COVID-19 [142]. Moreover, increasing data indicate new means of treatment, targeting the above proteins with a specific inhibitor (ICI), with the best effect achieved in combination with other drugs, such as kinase inhibitors in cancer therapy, or CAR-T cells in lymphomas and leukemias [143]. In recent months, numerous studies have proved that these strategies can also be used against SARS-CoV-2 infection and new studies are currently under clinical investigation [144]. 

The most important chemokines and cytokines were mentioned above. Elevated levels of reactive oxygen species (ROS) worsen prognosis in patients with cardiovascular comorbidities [145]. SARS-CoV-2 S protein also has an impact on hematopoietic and progenitor stem cells, causing hematological disorders (e.g., thrombocytopenia, lymphocytopenia) [146]. In patients with severe COVID-19, high concentrations of numerous cytokines, chemokines, and eicosanoids have been found [147]. Molecular and epigenetic mechanisms that regulate the pathogenesis of COVID-19 are complex processes (e.g., DNA methylation and oxidation, histone modifications/chromatin remodeling, non-coding RNAs), responsible for moderating the innate response during infection [148]. Interestingly, non-immune cells, which are present in pulmonary airways, also play an important role in the pathogenesis of COVID-19. Autophagy, which can be increased or decreased, depends on various cytokines [149]. All of these factors are important parts of the pathogenesis of COVID-19 (Figure 3).

## Figures and Tables

**Figure 1 cells-10-01817-f001:**
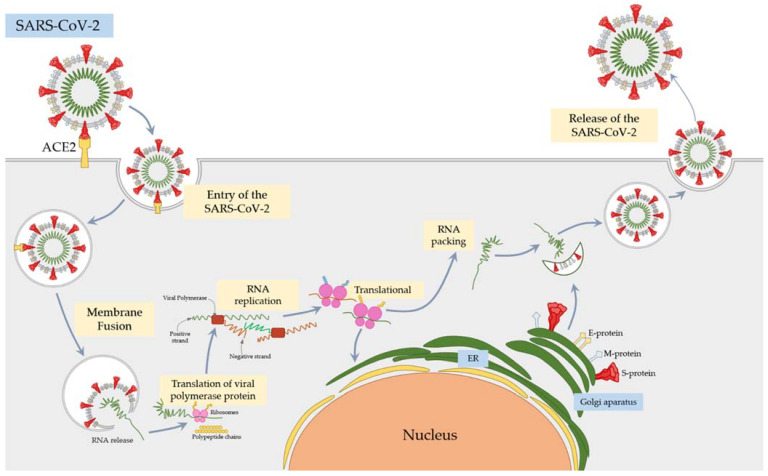
The schematic diagram of the mechanism of SARS-CoV-2 entry, viral replication, and viral RNA packing in the cell.

**Figure 2 cells-10-01817-f002:**
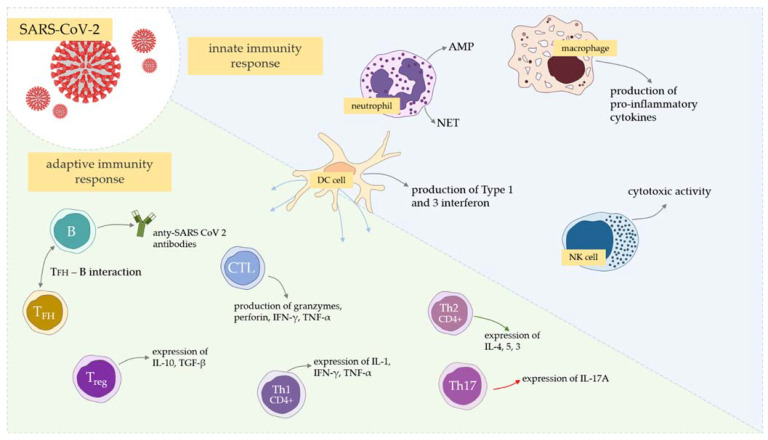
The summary of innate and adaptive responses after infection with SARS-CoV-2.

**Figure 3 cells-10-01817-f003:**
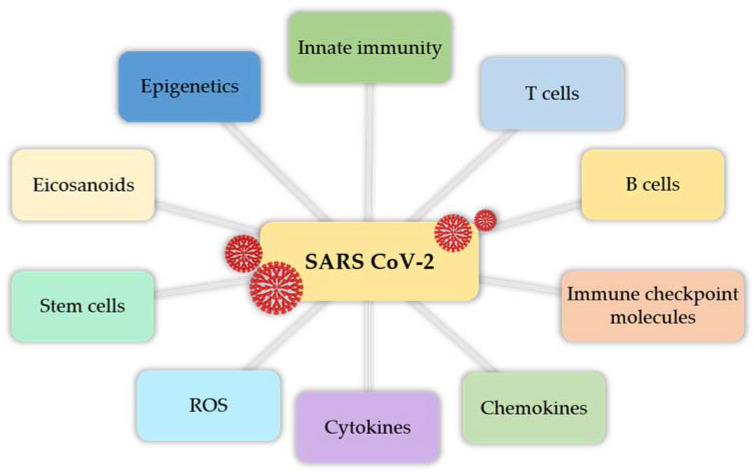
A schematic graph showing numerous factors involved in immune response to SARS-CoV-2 infection.

**Table 1 cells-10-01817-t001:** The comparison of information at the beginning of the COVID-19 pandemic with the updated information obtained during research on SARS-CoV-2.

	Beginning of the Pandemic	Current Knowledge
**Main line of defense**	Acquired immunity	Innate immunity
**Response in leukocytes**	Lymphopenia, neutropenia	NETs, depletion of CD8+ and CD4+
**Diagnosis**	PCR, antigen, antibodies	Specific T cells, B cells, NK
**Prognostic factors**	LDH, cytokines (e.g., IL-6, IL-10)	Immune Checkpoint Molecules **
**Treatment options**	Remdesivir, Tocilizumab, GCs *	Immune Checkpoint Inhibitors ***

* Glucocorticosteroids; ** PD-1, CTLA-4, TIM-3, LAG-3, TIGIT; *** anti-PD-1, anti-PD-L1, anti-CTLA-4, anti-TIM-3, anti-LAG-3, anti-TIGIT.

## Data Availability

Not applicable.

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
