# Peer review of "Interplay between Neutrophils, NETs and T-Cells in SARS-CoV-2 Infection—A Missing Piece of the Puzzle in the COVID-19 Pathogenesis?"

_cells, 2021, doi:10.3390/cells10071817_

Round 1

Reviewer 1 Report

The paper by Niedzwiedzka-Rystwej et al. reports the most important informations actually present in leterature to understand the status of immune system in Sars-Cov-2 infection.

In this review, the authors well explain the characteristics of innate and adaptive immune response during Sars-Cov-2 infection. They explain the role of neutrophils in COVID-19 and their ability to regulate cytokines activity and also the role of lymphocites.

Niedzwiedzka-Rystwej et al. try to explain the severity of clinical disease with the characteristics of individual immune response.

All this features are important to achieve a better diagnostic, therapeutic and prophylactic methods for the future.

The paper is well designed and written. It can be accepted in this form.

Author Response

Dear Reviewer,

On behalf of the Authors, I would like to thank you for the informative and prompt suggestion to our article entitled: “Rarely described elements of immunity against SARS-CoV-2 – a missing puzzle in the COVID-19 pathogenesis?” written by Paulina Niedźwiedzka-Rystwej, Ewelina Grywalska, Rafał Hrynkiewicz, Dominika Bębnowska, Mikołaj Wołącewicz, Adam Majchrzak and Miłosz Parczewski.  

We have revised the manuscript according to the suggestion, and here are point-by point answers.

Reviewer 1

The paper by Niedzwiedzka-Rystwej et al. reports the most important informations actually present in leterature to understand the status of immune system in Sars-Cov-2 infection.

In this review, the authors well explain the characteristics of innate and adaptive immune response during Sars-Cov-2 infection. They explain the role of neutrophils in COVID-19 and their ability to regulate cytokines activity and also the role of lymphocites.

Niedzwiedzka-Rystwej et al. try to explain the severity of clinical disease with the characteristics of individual immune response.

All this features are important to achieve a better diagnostic, therapeutic and prophylactic methods for the future.

The paper is well designed and written. It can be accepted in this form.

RE: We would like to thank the Reviewer for such a positive feedback on our article.

Yours sincerely,

Paulina Niedźwiedzka-Rystwej

Reviewer 2 Report

Paulina Niedźwiedzka-Rystwej present a very interesting overview of the immuno-pathological analysis of COVID 19 caused by SARS COV2 virus. 

After careful reading of the review , I have following concerns which need to be addressed by the authors 

  1. Although this review focus around the role of NET and T cells but lack the macrophages as one of the major driver of ARDS and death of the severely infected patients.
  2. Which epitopes of SARS COV 2 interact with neutrophils and macrophage seperatly or they can interact with these cells simultaneously 
  3. What actually drive Th2/ 17 programming of infected patients. what are the mechanisms 
  4. Neutrophiles does frequently and macrophage comes and pick up these cells for clearance . whether this contribute to the Th2 programming of the pulmonary compartment . 
  5. Which trigger Macrophage activation syndrome - is not discussed 
  6. In view of several key papers addressing the significance  of macrophages activation syndrome, I would advice author to include information related to macrophages and their significance.
  7. Based the current state of art and information, please include one table which address the putative cellular targets for reconditioning of immune cells of the patients .

Author Response

Dear Reviewer,

On behalf of the Authors, I would like to thank you for the informative and prompt suggestion to our article entitled: “Rarely described elements of immunity against SARS-CoV-2 – a missing puzzle in the COVID-19 pathogenesis?” written by Paulina Niedźwiedzka-Rystwej, Ewelina Grywalska, Rafał Hrynkiewicz, Dominika Bębnowska, Mikołaj Wołącewicz, Adam Majchrzak and Miłosz Parczewski.  

We have revised the manuscript according to the suggestions, and here are point-by point answers.

Paulina Niedźwiedzka-Rystwej present a very interesting overview of the immuno-pathological analysis of COVID 19 caused by SARS COV2 virus. 

After careful reading of the review , I have following concerns which need to be addressed by the authors 

  1. Although this review focus around the role of NET and T cells but lack the macrophages as one of the major driver of ARDS and death of the severely infected patients.

RE: We have used the kind suggestion of the Reviewer and added data on macrophages (additions marked in red)

  1. Which epitopes of SARS COV 2 interact with neutrophils and macrophage seperatly or they can interact with these cells simultaneously 

RE: Unfortunately, we are not able to dispel this doubt, because there is not much information on this issue in the literature, or we are not able to reach it. However, this content is indeed interesting to consider.

  1. What actually drive Th2/ 17 programming of infected patients. what are the mechanisms 

RE: We were actually able to information only on Th17 programming.

  1. Neutrophiles does frequently and macrophage comes and pick up these cells for clearance . whether this contribute to the Th2 programming of the pulmonary compartment. 

RE: Unfortunately, we do not have the knowledge to answer this question.

  1. Which trigger Macrophage activation syndrome - is not discussed 

RE: We have discussed the issue of macrophages activation in the current form of the manuscript.

  1. In view of several key papers addressing the significance  of macrophages activation syndrome, I would advice author to include information related to macrophages and their significance.

RE: We expanded on the macrophages activation syndrome in the current form of the macrophages.

  1. Based the current state of art and information, please include one table which address the putative cellular targets for reconditioning of immune cells of the patients.

RE: We believe that the mentioned table does not match our article. However, thank you for the suggestion.

Thank you once again for time and consideration. We do hope that now the manuscript after revision will fulfill the requirements.

Yours sincerely,

Paulina Niedźwiedzka-Rystwej

Reviewer 3 Report

While the authors claimed that this review would focus on "rarely described elements of the immune system", the two elements, the roles of neutrophils and NETs and depletion of T lymphocytes in COVID-19, have actually been heavily reviewed. A quick PubMed search on "COVID-19 and neutrophils" yielded 1,424 results, including 152 reviews. There have been good reviews and meta-analysis on both elements. For example:

Borges L, Pithon-Curi TC, Curi R, Hatanaka E. COVID-19 and Neutrophils: The Relationship between Hyperinflammation and Neutrophil Extracellular Traps. Mediators Inflamm. 2020 Dec 2;2020:8829674. doi: 10.1155/2020/8829674. PMID: 33343232; PMCID: PMC7732408.

Gustine JN, Jones D. Immunopathology of Hyperinflammation in COVID-19. Am J Pathol. 2021 Jan;191(1):4-17. doi: 10.1016/j.ajpath.2020.08.009. Epub 2020 Sep 11. PMID: 32919977; PMCID: PMC7484812.

Huang W, Berube J, McNamara M, Saksena S, Hartman M, Arshad T, Bornheimer SJ, O'Gorman M. Lymphocyte Subset Counts in COVID-19 Patients: A Meta-Analysis. Cytometry A. 2020 Aug;97(8):772-776. doi: 10.1002/cyto.a.24172. Epub 2020 Jul 18. PMID: 32542842; PMCID: PMC7323417.

None of these important articles have been cited by the authors. This manuscript did not provide any new information as they claimed in their manuscript. Even targeting NET and T lymphocytes to treat COVID-19 has been published:

Barnes BJ, Adrover JM, Baxter-Stoltzfus A, Borczuk A, Cools-Lartigue J, Crawford JM, Daßler-Plenker J, Guerci P, Huynh C, Knight JS, Loda M, Looney MR, McAllister F, Rayes R, Renaud S, Rousseau S, Salvatore S, Schwartz RE, Spicer JD, Yost CC, Weber A, Zuo Y, Egeblad M. Targeting potential drivers of COVID-19: Neutrophil extracellular traps. J Exp Med. 2020 Jun 1;217(6):e20200652. doi: 10.1084/jem.20200652. PMID: 32302401; PMCID: PMC7161085.

Other issues: the authors described some generic functions in the immune responses, but not very specific to COVID-19. The summaries and analyses of literature are insufficient and unclear. The whole section 2 in the manuscript is irrelevant to the main focus of the manuscript, it should be left out from the manuscript.

It is not recommended that this manuscript be accepted.

Author Response

Dear Reviewer,

On behalf of the Authors, I would like to thank you for the informative and prompt suggestion to our article entitled: “Rarely described elements of immunity against SARS-CoV-2 – a missing puzzle in the COVID-19 pathogenesis?” written by Paulina Niedźwiedzka-Rystwej, Ewelina Grywalska, Rafał Hrynkiewicz, Dominika Bębnowska, Mikołaj Wołącewicz, Adam Majchrzak and Miłosz Parczewski.  

We have revised the manuscript according to the suggestion, and here are point-by point answers.

  1. While the authors claimed that this review would focus on "rarely described elements of the immune system", the two elements, the roles of neutrophils and NETs and depletion of T lymphocytes in COVID-19, have actually been heavily reviewed. A quick PubMed search on "COVID-19 and neutrophils" yielded 1,424 results, including 152 reviews. There have been good reviews and meta-analysis on both elements. For example:

Borges L, Pithon-Curi TC, Curi R, Hatanaka E. COVID-19 and Neutrophils: The Relationship between Hyperinflammation and Neutrophil Extracellular Traps. Mediators Inflamm. 2020 Dec 2;2020:8829674. doi: 10.1155/2020/8829674. PMID: 33343232; PMCID: PMC7732408.

Gustine JN, Jones D. Immunopathology of Hyperinflammation in COVID-19. Am J Pathol. 2021 Jan;191(1):4-17. doi: 10.1016/j.ajpath.2020.08.009. Epub 2020 Sep 11. PMID: 32919977; PMCID: PMC7484812.

Huang W, Berube J, McNamara M, Saksena S, Hartman M, Arshad T, Bornheimer SJ, O'Gorman M. Lymphocyte Subset Counts in COVID-19 Patients: A Meta-Analysis. Cytometry A. 2020 Aug;97(8):772-776. doi: 10.1002/cyto.a.24172. Epub 2020 Jul 18. PMID: 32542842; PMCID: PMC7323417.

RE: The manuscript bibliography has been updated with the items proposed by the Reviewer and other recent original publications that bring new knowledge to the scientific world about the role of neutrophils, NETs and T lymphocytes in the course of COVID-19, so that the article actually compiles the latest literature. However, all the works mentioned by the reviewer focus on single cell types of the immune system, while the manuscript describes and combines the role of both cell types in the course and pathogenesis of COVID-19.

  1. None of these important articles have been cited by the authors. This manuscript did not provide any new information as they claimed in their manuscript. Even targeting NET and T lymphocytes to treat COVID-19 has been published:

Barnes BJ, Adrover JM, Baxter-Stoltzfus A, Borczuk A, Cools-Lartigue J, Crawford JM, Daßler-Plenker J, Guerci P, Huynh C, Knight JS, Loda M, Looney MR, McAllister F, Rayes R, Renaud S, Rousseau S, Salvatore S, Schwartz RE, Spicer JD, Yost CC, Weber A, Zuo Y, Egeblad M. Targeting potential drivers of COVID-19: Neutrophil extracellular traps. J Exp Med. 2020 Jun 1;217(6):e20200652. doi: 10.1084/jem.20200652. PMID: 32302401; PMCID: PMC7161085.

RE: The manuscript has been updated with the reference indicated by the Reviewer. However, this work concerns only NETs, while the discussed manuscript also touches upon the subject of T lymphocytes, and collects the knowledge available at the moment in both topics.

  1. Other issues: the authors described some generic functions in the immune responses, but not very specific to COVID-19. The summaries and analyses of literature are insufficient and unclear. The whole section 2 in the manuscript is irrelevant to the main focus of the manuscript, it should be left out from the manuscript.

It is not recommended that this manuscript be accepted.

RE: Section 2 of this manuscript is according to the Authors, an important element of this review, and we would prefer to keep it. It introduces the subject of COVID-19 pathogenesis, describes both molecular and structural mechanisms of the course of the disease, as well as, most importantly, potential development directions in the context of disease treatment. These potential development strategies are then confronted with the knowledge available about the roles of the immune cells discussed in the manuscript.

Thank you once again for time and consideration. We do hope that now the manuscript after revision will fulfill the requirements.

Yours sincerely,

Paulina Niedźwiedzka-Rystwej

Reviewer 4 Report

The manuscript titled “Rarely described elements of immunity against SARS-CoV-2 – 2 a missing puzzle in the COVID-19 pathogenesis?” is well planned and carefully written piece of work. The manuscript describes some rare elements of immunity to potentially understand the pathogenesis of COVID-19 and some treatment options. This manuscript when publish will interest ″Cells″ readerships as well as the scientific community and the world at large.  Some improvements are needed to enhance its clarity. Also, there are grammar issues that need to be fixed. 

Below are some suggestions 

Line 36 Please consider deleting “or novel coronavirus (2019-nCoV)” and replace with an agent causing the current corona disease 19 (COVID-19) 
Line 46 …..  safe and effective preventive measures such as????? Please mention some
Line  47 starting from “but” please fix that sentence 
Line 50 please delete understanding after “as well as”
Line 51 please consider replacing “in the course of” with 
Line 53 to 56 which authors and papers are you referring to? Please cite them
Line 59 It was reported…not found. Please correct this
Line 60 to 61It should go…………………….antagonist against type I interferons which inhibits  innate mechanisms 
Line 63 to 64 ………….This phenomenon triggers ARDS. Also references is need here 

Line 91 to 91 (as of March 21, 2021,). Please update this to June, 2021. Also has spread not is spreading 
Line 85 to 104 Please cite Seyran et al. Questions concerning the proximal origin of SARS‐CoV‐2 J Med Virol. 2021 Mar;93(3):1204-1206. doi: 10.1002/jmv.26478.

Line 119 to 119 please delete not relevant 
Line 146 to 163 importance literature is missing here : The structural basis of accelerated host cell entry by SARS-CoV-2: https://doi.org/10.1111/febs.15651; Possible Transmission Flow of SARS-CoV-2 Based on ACE2 Features. Molecules 2020, 25, 5906; doi:10.3390/molecules25245906

Line 167. This section needs further improvement  
Line 331 “Studies have also shown that the severe course of COVID-19”….which studies ??????please state

Author Response

Dear Reviewer,

On behalf of the Authors, I would like to thank the Reviewers for the informative and prompt suggestion to our article entitled: “Rarely described elements of immunity against SARS-CoV-2 – a missing puzzle in the COVID-19 pathogenesis?” written by Paulina Niedźwiedzka-Rystwej, Ewelina Grywalska, Rafał Hrynkiewicz, Dominika Bębnowska, Mikołaj Wołącewicz, Adam Majchrzak and Miłosz Parczewski.  

We have revised the manuscript according to the suggestion, and here are point-by point answers.

The manuscript titled “Rarely described elements of immunity against SARS-CoV-2 – 2 a missing puzzle in the COVID-19 pathogenesis?” is well planned and carefully written piece of work. The manuscript describes some rare elements of immunity to potentially understand the pathogenesis of COVID-19 and some treatment options. This manuscript when publish will interest ″Cells″ readerships as well as the scientific community and the world at large.  Some improvements are needed to enhance its clarity. Also, there are grammar issues that need to be fixed. 

Below are some suggestions 

  1. Line 36 Please consider deleting “or novel coronavirus (2019-nCoV)” and replace with an agent causing the current corona disease 19 (COVID-19).

RE: As suggested by the reviewer, we decided to delete “or novel coronavirus (2019-nCoV)” and replaced with an agent causing the current corona disease 19 (COVID-19). Thank you (currently Line 36).

  1. Line 46 …..  safe and effective preventive measures such as????? Please mention some

RE: Thank you for your valuable attention. As suggested by the reviewer, we have listed safe and effective measures to prevent SARS-CoV-2 transmission (currently Line 47 to 50).

  1. Line  47 starting from “but” please fix that sentence 

RE: Thank you for your valuable attention. We corrected the sentence as suggested by the reviewer.

  1. Line 50 please delete understanding after “as well as”

RE: The sentence was corrected as suggested by the reviewer. Thank you (currently Line 54 to 55).

  1. Line 51 please consider replacing “in the course of” with 

RE: The sentence was corrected as suggested by the reviewer. Thank you (currently Line 55).

  1. Line 53 to 56 which authors and papers are you referring to? Please cite them

RE: As suggested by the reviewer, we cited the authors. Thank you (currently Line 58).

  1. Line 59 It was reported…not found. Please correct this

RE: The sentence was corrected as suggested by the reviewer. Thank you (currently Line 63).

  1. Line 60 to 61It should go…………………….antagonist against type I interferons which inhibits  innate mechanisms 

RE: The sentence was corrected as suggested by the reviewer. Thank you (currently Line 64 to 65).

  1. Line 63 to 64 ………….This phenomenon triggers ARDS. Also references is need here 

RE: Thank you for your valuable attention. We have added references.

  1. Line 91 to 91 (as of March 21, 2021,). Please update this to June, 2021. Also has spread not is spreading

RE: We corrected the date. Thank you for your vigilance (currently Line 131).

  1. Line 85 to 104 Please cite Seyran et al. Questions concerning the proximal origin of SARS‐CoV‐2 J Med Virol. 2021 Mar;93(3):1204-1206. doi: 10.1002/jmv.26478.

RE: We have cited the suggested paper.

  1. Line 119 to 119 please delete not relevant

RE: The sentence was corrected as suggested by the reviewer. Thank you.

  1. Line 146 to 163 importance literature is missing here : The structural basis of accelerated host cell entry by SARS-CoV-2: https://doi.org/10.1111/febs.15651; Possible Transmission Flow of SARS-CoV-2 Based on ACE2 Features. Molecules 2020, 25, 5906; doi:10.3390/molecules25245906

RE: Thank you for the suggestion, we have cited the missing literature, according to the Reviewers suggestion.

  1. Line 167. This section needs further improvement

RE: Thank you very much for your valuable attention. After seeing all the reviews and after internal consultation, we decided to remove section 2.3 from this review.

  1. Line 331 “Studies have also shown that the severe course of COVID-19”….which studies ??????please state

RE: Thank you for your valuable attention. We have added references.

Thank you once again for time and consideration. We do hope that now the manuscript after revision will fulfill the requirements.

Yours sincerely,

Paulina Niedźwiedzka-Rystwej

Reviewer 5 Report

All the discussion has been around what is already known about COVID-19. The authors need to give a table giving what is already known and what is not known and what is new ad what the authors propose apart from what is known.

I find that there is very little new information  in the review. So the authors need to outline what they think is new in their review compared to what is already known. 

There are so many variables from person to person in terms of their innate and adaptive immune responses, factors secreted by them, and the quality and quantity of these factors, and the duration of their presence in the plasma and tissues and even the response of non-immune tissues to the virus. Possible interaction between immune and non-immune cells in response to SARS-CoV-2 need to be discussed and interpreted in term of response of patients to the disease (recovery and severity of illness). 

All the issues need to be discussed in a comprehensive and cohesive fashion. 

I think emphasis should be placed on cytokines, ROS, anti-oxidants, stem cells, chemokines, eicosanoids, and reaction of immune and non-immune cells response to SARS-CoV-2 and of course the dose of virus in the initial stage of infection.

Some variations between individuals could be related to the dose of viruses exposed, general nutritional status of the subject, atmospheric pollution, other co-morbidity conditions and their impact on the COVID-19. All these factors are important and need consideration.      

Author Response

Dear Reviewer,

On behalf of the Authors, I would like to thank you for the informative and prompt suggestion to our article entitled: “Rarely described elements of immunity against SARS-CoV-2 – a missing puzzle in the COVID-19 pathogenesis?” written by Paulina Niedźwiedzka-Rystwej, Ewelina Grywalska, Rafał Hrynkiewicz, Dominika Bębnowska, Mikołaj Wołącewicz, Adam Majchrzak and Miłosz Parczewski.  

We have revised the manuscript according to the suggestion, and here are point-by point answers.

All the discussion has been around what is already known about COVID-19. The authors need to give a table giving what is already known and what is not known and what is new ad what the authors propose apart from what is known.

  1. I find that there is very little new information in the review. So the authors need to outline what they think is new in their review compared to what is already known. 

RE: We tried to emphasize the importance of the novel elements of immunity throughout the paper. According to your advice, we added new table, that compares them to what is already known. Nevertheless, our point in this review was to summarize the current knowledge, underlining what new has been recently described, rather than present completely unknown state of the facts, what we plan to present in another study.

  1. There are so many variables from person to person in terms of their innate and adaptive immune responses, factors secreted by them, and the quality and quantity of these factors, and the duration of their presence in the plasma and tissues and even the response of non-immune tissues to the virus. Possible interaction between immune and non-immune cells in response to SARS-CoV-2 need to be discussed and interpreted in term of response of patients to the disease (recovery and severity of illness).

RE: Thank you for your answer with possible interactions between immune and non-immune cells in COVID-19. However, our review treats the immunological mechanisms of response to the disease, that’s why we didn’t focus on other factors. According to your precious advice, we added some of these facts in our review.

  1. All the issues need to be discussed in a comprehensive and cohesive fashion. 

RE: We tried to rewrite the content of the manuscript to make it more comprehensive.

  1. I think emphasis should be placed on cytokines, ROS, anti-oxidants, stem cells, chemokines, eicosanoids, and reaction of immune and non-immune cells response to SARS-CoV-2 and of course the dose of virus in the initial stage of infection.

RE: Apart from our deep gratitude for your comment, our review was not dedicated to tract about non-immunological mechanism of response to SARS-CoV-2, like these mentioned in above statement. Notwithstanding, we expanded our review with more information about mentioned factors and their potential impact on patient’s reaction to COVID-19. To make the facts clearer, we have added a Figure summarizing the role of the mentioned elements.

  1. Some variations between individuals could be related to the dose of viruses exposed, general nutritional status of the subject, atmospheric pollution, other co-morbidity conditions and their impact on the COVID-19. All these factors are important and need consideration.  

RE: Thank you again for the opinion. Indeed, multifactorial analysis of all mentioned, individual aspects would be a perfect tool of complex diagnosis. We will consider to describe it in a next study.

Thank you once again for time and consideration. We do hope that now the manuscript after revision will fulfill the requirements.

Yours sincerely,

Paulina Niedźwiedzka-Rystwej

Round 2

Reviewer 2 Report

The manuscript has sufficient information to be published !!

However some language issue need to be corrected at proof stage 

Author Response

Dear Reviewer,

On behalf of the Authors, I would like to thank you for accepting the manuscript in the corrected form. We would also like to inform you, that the manuscript has undergone an English editing, as you kindly suggested.

Once again, thank you for your time and consideration.

Regards,

Paulina Niedźwiedzka-Rystwej

Reviewer 3 Report

The manuscript seemed to be improved over the previous version. It still requires significant improvement to be considered for publication.

  1. The most important thing, revise the titile, abstract, and introduction to emphasize on the main points of the manuscript, i.e. neutrophiles and NETs and subset of T cell and T cell depletion in COVID-19 pathogenesis, rather than "rarely described elements of immunity", as these are really not rarely described and immune responses have been the focus on COVID-19 research since the start of the pandemic.
  2. The author should make connections between the two elements, NETs and T cell exhaustion in the immune responses to SARS-CoV-2. The authors claimed they are looking into the combined roles of these two immune elements, but these two elements were described separately and the authors made no connections between them. There are multiple factors affecting COVID-19 progresses, and the authors chose these two types of immune responses in this manuscript. It needs more rationales than simply "rarely described". It actually reflected in Figure 2, where it appears that innate and adaptive immune responses are separates, while it should emphasize the linkage between the two (more than just dendritic cells) and the combined roles in COVID.
  3. Table 1 is kind of confusing. It requires more explanation - what did the authors mean by "new knowledge", how does this "new knowledge" hlep in understanding and fighting the pandemic, why did the authors think the listed items are "new knowledge". What consists "new knowledge" is really dependent on who the readers are. Also specify the "immune checkpoint inhibitors" and "molecules". Figure 3 also requires explanation and clarification in one paragrah, even thought the items on the figure are not the focus of this manuscript.

Author Response

Dear Reviewer,

We would like to deeply thank you for you time and effort you have devoted in reading and making ur paper a better one. Please find below our responses to the issues raised by you:

  1. The most important thing, revise the titile, abstract, and introduction to emphasize on the main points of the manuscript, i.e. neutrophiles and NETs and subset of T cell and T cell depletion in COVID-19 pathogenesis, rather than "rarely described elements of immunity", as these are really not rarely described and immune responses have been the focus on COVID-19 research since the start of the pandemic.

RE: According to the suggestion of the Reviewer, we have changed the title and abstract to omit the wrongly used statement “rarely described”. We do agree with the Reviewer that every day shows that those elements are not rarely described. Thank you for your kind suggestion.

  1. The author should make connections between the two elements, NETs and T cell exhaustion in the immune responses to SARS-CoV-2. The authors claimed they are looking into the combined roles of these two immune elements, but these two elements were described separately and the authors made no connections between them. There are multiple factors affecting COVID-19 progresses, and the authors chose these two types of immune responses in this manuscript. It needs more rationales than simply "rarely described". It actually reflected in Figure 2, where it appears that innate and adaptive immune responses are separates, while it should emphasize the linkage between the two (more than just dendritic cells) and the combined roles in COVID.

RE: We have tried to bond the elements of the immune system described in the paper with adding more explanation in the section 5. We hope this may fullfill the greatly appreciated idea of the Reviewer.

  1. Table 1 is kind of confusing. It requires more explanation - what did the authors mean by "new knowledge", how does this "new knowledge" hlep in understanding and fighting the pandemic, why did the authors think the listed items are "new knowledge". What consists "new knowledge" is really dependent on who the readers are. Also specify the "immune checkpoint inhibitors" and "molecules". Figure 3 also requires explanation and clarification in one paragrah, even thought the items on the figure are not the focus of this manuscript.

RE: We have corrected the Table 1 to reduce confusion and described some elements shown in the table in more details. We have left the Figure 3 in place, as it was requested by a different Reviewer.

Once again thank you very much and we hope that the paper will fulfill the requirements in the present form.

Regards,

Paulina Niedźwiedzka-Rystwej

Reviewer 5 Report

accept

Author Response

Dear Reviewer,

Thank you for the acceptance.

Best regards,

Paulina Niedźwiedzka-Rystwej